# The Multifaceted Effects of Short-Term Acute Hypoxia Stress: Insights into the Tolerance Mechanism of *Propsilocerus akamusi* (Diptera: Chironomidae)

**DOI:** 10.3390/insects14100800

**Published:** 2023-10-03

**Authors:** Yao Zhang, Qing-Ji Zhang, Wen-Bin Xu, Wei Zou, Xian-Ling Xiang, Zhi-Jun Gong, Yong-Jiu Cai

**Affiliations:** 1Key Laboratory of Watershed Geographic Sciences, Nanjing Institute of Geography and Limnology, Chinese Academy of Sciences, Nanjing 210008, China; zy0000@ahnu.edu.cn (Y.Z.); wzou@niglas.ac.cn (W.Z.); zjgong@niglas.ac.cn (Z.-J.G.); 2School of Ecology and Environment, Anhui Normal University, Wuhu 241002, China; xlxiang@ahnu.edu.cn; 3Collaborative Innovation Center of Recovery and Reconstruction of Degraded Ecosystem in Wanjiang Basin Co-founded by Anhui Province and Ministry of Education, Wuhu 241002, China; 4School of Geography and Ocean Science, Nanjing University, Nanjing 210023, China; zhangqj@smail.nju.edu.cn; 5College of Animal Sciences, Zhejiang University, Hangzhou 310058, China; xuwenbin9143@zju.edu.cn

**Keywords:** Chironomidae, acute hypoxic stress, multi-omics analysis, energy metabolism, antioxidant mechanism

## Abstract

**Simple Summary:**

Eutrophication and global warming have caused acute hypoxia in aquatic ecosystems. However, *Propsilocerus akamusi* depends on great hypoxia tolerance to become a dominant species in eutrophic lakes, but the mechanism of this hypoxia tolerance is unclear. Thus, we combined physiological indicators and histomorphology observations with metabolome–transcriptome analysis to explore the mechanism comprehensively. The results showed that hypoxia tolerance mainly relies on apoptosis, energy metabolism, and an antioxidant mechanism. *P. akamusi* derives its energy from glycogen metabolism, lipid metabolism, protein digestion and absorption, and the glyoxydate cycle. Lactate is the end product of glycogen degradation, and HIF-1 plays an important role in promoting glycogenolysis in acute hypoxic conditions. However, ethanol probably originates from symbiodinium and, together with hydrogen peroxide, stimulates the elevation of catalase (CAT) activity and induced apoptosis. Understanding the processes that enable *P. akamusi* to survive lengthy periods of hypoxia in eutrophic lakes might provide a scientific reference for assessing toxicity and favoring policies to reduce their impact on the environment.

**Abstract:**

Plenty of freshwater species, especially macroinvertebrates that are essential to the provision of numerous ecosystem functions, encounter higher mortality due to acute hypoxia. However, within the family Chironomidae, a wide range of tolerance to hypoxia/anoxia is displayed. *Propsilocerus akamusi* depends on this great tolerance to become a dominant species in eutrophic lakes. To further understand how *P. akamusi* responds to acute hypoxic stress, we used multi-omics analysis in combination with histomorphological characteristics and physiological indicators. Thus, we set up two groups—a control group (DO 8.4 mg/L) and a hypoxic group (DO 0.39 mg/L)—to evaluate enzyme activity and the transcriptome, metabolome, and histomorphological characteristics. With blue–black chromatin, cell tightness, cell membrane invagination, and the production of apoptotic vesicles, tissue cells displayed typical apoptotic features in the hypoxic group. Although lactate dehydrogenase (LDH), alcohol dehydrogenase (ADH), catalase (CAT), and Na+/K+ -ATPase (NKA) activities were dramatically enhanced under hypoxic stress, glycogen content, and superoxide dismutase (SOD) activities were significantly reduced compared to the control group. The combined analysis of the transcriptome and metabolome, which further demonstrated, in addition to carbohydrates, including glycogen, the involvement of energy metabolism pathways, including fatty acid, protein, trehalose, and glyoxylate cycles, provided additional support for the aforementioned findings. Lactate is the end product of glycogen degradation, and HIF-1 plays an important role in promoting glycogenolysis in acute hypoxic conditions. However, we discovered that the ethanol tested under hypoxic stress likely originates from the symbiodinium of *P. akamusi*. These results imply that some parameters related to energy metabolism, antioxidant enzyme activities, and histomorphological features may be used as biomarkers of eutrophic lakes in *Chironomus riparius* larvae. The study also provides a scientific reference for assessing toxicity and favoring policies to reduce their impact on the environment.

## 1. Introduction

Hypoxic zones have been rapidly expanding in both space and frequency over the past two decades in freshwater ecosystems as a result of rising nutrient loadings and global warming [1,2,3,4]. However, acute hypoxia leads to higher mortality rates in macroinvertebrates, which play a supporting role in the material cycle and energy flow in freshwater ecosystems [5,6]. The composition of the macroinvertebrate community has drastically changed due to hypoxia, which has also altered the metabolic response (metabolomes) and perhaps other biochemical processes, such as the level of oxidative stress [7,8].

Hypoxic stress alters the metabolite content of insects relating to energy and antioxidants [9,10]. Hypoxic stress increases reactive oxygen species (ROS) production in the body by disrupting the electron transport chain and the antioxidant defense system, including enzymes such as superoxide dismutase (SOD) and catalase (CAT), which is activated to avoid the harmful effects of ROS on biomolecules [11,12]. For instance, in the aquatic insect *Belostoma elegans*, hypoxic conditions have been shown by Lavarias et al. (2017) to be one of the factors affecting antioxidant enzyme activity [13]. The evidence indicates that ethanol is the only byproduct of glycogen degradation from *Chironomus thummi* and *Culex pipiens* in hypoxic stress, and it is catalyzed by alcohol dehydrogenase (ADH) or diffused into the surrounding water [14,15]. However, a study on *Chaoborus crystallinus* found that large quantities of alanine are accumulated, while lactate is produced as a minor end product under anoxic conditions [16]. In addition, hypoxic stress not only led to significant tissue damage but also induced cell apoptosis, and the expression of hypoxia-inducible genes has been identified in bivalve and crustacea [17,18,19,20,21,22,23,24]. Studies have also shown in scallops, crabs, and clams that energy-consuming processes (including protein turnover and ion transport) are suppressed by hypoxic stress, which is indicated by a dramatic decrease in the activity of Na+/K+ -ATPase (NKA) [25,26]. However, the effects of hypoxia on freshwater macroinvertebrates are unclear, especially the family Chironomidae, which possesses a wide range of tolerances to hypoxia [27,28,29].

Chironomids are widely dispersed, simple to cultivate, have brief life cycles, and are also adapted to a variety of environmental rigors, including desiccation, anoxia, high temperature, freezing, eutrophication, and chemical pollution [30]. Thus, they are a suitable taxon to explore the adaptation mechanisms required to endure environmental stress, to monitor water quality as bioindicators, and to test the toxicity of chemicals in ecotoxicological assays [30,31,32]. Chironomid midge larvae possess a wide range of tolerances to hypoxia due to extracellular hemoglobins (Hbs) in monomeric and dimeric forms floating in their hemolymph [28,33]. In particular, *P. akamusi* of Chironomidae demonstrated incredibly low fuel consumption at a stage of estivation when the larvae lived in deep sediment and engaged in anaerobic respiration, with ethanol as a major metabolite, to withstand prolonged hypoxia. This may be an important reason the species frequently predominates macroinvertebrates in many eutrophic lakes [34,35,36]. However, the mechanisms of how *P. akamusi* survives for a prolonged period of time in acute hypoxic environments are still unclear.

Approaches to *P. akamusi* have mostly focused on the transcriptome, proteome, or metabolome [37,38,39,40,41]. However, single omics cannot systematically explain the rapid biochemical reactions and metabolic changes that are expected for acute hypoxia [42,43]. For this reason, we conducted a comprehensive comparative metabolome–transcriptome analysis to understand how *P. akamusi*, the dominant macroinvertebrate species in many eutrophic lakes, tolerates prolonged hypoxia [44]. Thus, we combined physiological indicators and histomorphology observations with metabolome–transcriptome analysis to comprehensively assess the histomorphological features, energy metabolism mechanisms, and antioxidant mechanisms of *P. akamusi* larvae exposed to acute hypoxia. We predicted (1) that acute hypoxic stress induced apoptosis and significantly increased NKA activity in *P. akamusi* compared to controls. (2) Hypoxia caused a significant decrease in glycogen content, promoted a significant increase in ADH and LDA, and (3) also promoted a significant increase in SOD and CAT compared to the control group. (4) In addition, the transcriptome and metabolome analysis further confirmed and refined the above results.

## 2. Materials and Methods

### 2.1. Experimental Animals, Hypoxia Challenge, and Tissue Sample Preparation

The larvae of *P. akamusi* were collected with a Surber sampler (30 × 30 cm, 500 μm mesh) in the Taihu Lake basin (N 31°45′37′′, E 120°43′56′′; altitude of 4 m) in June 2019. The viable *P. akamusi* were reared on an artificial diet of fish flake food (Xiamen Mincheng Imp&Exp Co. Ltd., Xiamen, Fujian, China) in culture boxes containing 2 L of dechlorinated tap water and 1–2 cm of acid-washed sand under 25 ± 1 °C temperature conditions. After 3 d of domestication, robust, large, and color-coordinated 4th instar larvae were selected for the experiment. A hypoxic experimental group and a control group with four biological replicates each were established. Larvae were transferred to 250 mL glass beakers containing 200 mL of aerated dechlorinated tap water (90 Larvae per beaker). The temperature was kept constant at 24–25 °C throughout the experiment, and no feeding was performed during exposures. The oxygen concentration in the control group was not less than 8.0 mg/L. In hypoxic groups, DO was controlled by bubbling nitrogen gas through the water tanks, and the oxygen content was checked by a DO meter (DO-31 P, TOADKK, Tokyo, Japan) at all times to ensure that it did not exceed 0.5 mg/L. Two larvae in each biological replicate were selected and washed in ultrapure water, then fixed in glass vials containing 4% Paraformaldehyde Fix Solution for HE staining at 96 h. Six whole *P. akamusi* larvae were selected from each biological replicate for physiological indicators at 0 h, 12 h, 24 h, 48 h, 72 h, and 96 h. Six larvae in each biological replicate were selected for transcriptome and metabolome analyses at 96 h. These larvae were put into 1.5 mL EP tubes after measuring wet weight using an electronic balance, briefly submerged in liquid nitrogen for short-term storage, and then kept at −80 °C in an ultra-low-temperature refrigerator.

### 2.2. HE Staining

The *P. akamusi* larvae were washed in pre-cooled saline at 4 °C, and the tissue was then chopped into small pieces and stored on ice before being placed in 10% formaldehyde solution and fixed for one day and one night before being placed in ethanol for three hours to dehydrate it. After being sectioned and paraffin-embedded, the slides were dried on a staining rack for 20 min before being submerged in a solution of xylene and ethanol for 12 min. The slides were then dehydrated in ethanol for six minutes, dehydrated in xylene for 20 min, washed in xylene, stained in hematoxylin for two minutes, rinsed under low-flow water for four minutes, and then soaked in eosin solution for 20 s. The slides were then sealed with gum, dried, and examined under a biological microscope. After drying, the films were examined under a biological microscope [45].

### 2.3. Tissue Enzyme Activity and Glycogen Assays

The kits for the determination of glycogen content, ADH activity, LDH activity, CAT activity, SOD activity, and NKA activity were purchased from Nanjing Jiancheng Biological Company (Nanjing, China), and the enzyme activities and glycogen content of the hypoxic and control groups were determined using tissues as experimental samples. The specific operation procedure and calculation formula were referred to in the instruction manual.

### 2.4. Transcriptome Analysis

Utilizing the TRIzol reagent, which is available from Invitrogen in Vacaville, CA, USA, total RNA was isolated from *P. akamusi* larvae. Total RNA quantity and integrity were assessed using the Agilent Bioanalyzer 2100 equipment (Agilent, Santa Clara, CA, USA). Each sample was chosen to have 1.5 g of high-quality RNA for the investigation. At a high temperature, divalent cations break the mRNA into minute pieces. The mRNA Seq preparation sample kit (Illumina, San Diego, CA, USA) instructions were then followed to construct the final cDNA library utilizing reverse transcription. On the Illumina HiSeq platform (LC Sciences, Houston, TX, USA), paired-end DNA sequencing was carried out according to the vendor’s suggested technique. Clean data (clean reads) were chosen from the raw data (raw reads) to explore the patterns of gene expression in *P. akamusi* larvae under hypoxic stress, and then the sequence quality was checked using FastQC (https://fastqc.com/). With Trinity 2.4.0, clean data were then put together into a transcriptome for reference [46]. The “gene” sequence (also known as the Unigene) was chosen as the cluster’s longest transcript. The Basic Local Alignment Search Tool (BLAST) was used for transcriptome annotation, while the DIAMOND technique searched numerous databases with an E-value of 1 × e^−5^. We employed Blast2 GO with NR annotation for Gene Ontology (GO) annotation and the default settings of KASS for KEGG pathway analysis. The comparison of the hypoxic and control groups was conducted using an evaluation of differential gene expression. To measure the amount of Unigene expression, *Drosophila melanogaster* was used. The R package edgeR [47] was used to identify the differentially expressed Unigenes with statistical significance (*p*-value < 0.05). Using Perl scripts in R language, the DEGs were assessed for GO enrichment and KEGG pathway enrichment based on hypergeometric distributions. The raw data can be viewed in the SRA (Sequence Read Archive) database, number PRJNA972485.

### 2.5. Metabolome Analysis

Metabolites were used for further Liquid Chromatograph–Mass Spectrometer (LC-MS) system analysis [48]. To obtain baseline correction, maximum alignment, maximum detection, accurate masses, and normalized intensity of the peak, Agilent MSD Chemstation (version E.02.00.493) using the default settings was used [49]. Then, maximum acquisition and deconvolution were performed using the automated mass spectrometry deconvolution recognition system (AMDIS). The identification of metabolites was then completed by contrasting mass fragment patterns and retention times across several databases. Differentiating metabolites between the hypoxic and control groups were mapped to their corresponding metabolic pathways with the KEGG pathway-based MetPA online tool (http://metpa.metabolomics.ca/). Only metabolic pathways with −lg (*p*) < 1.301 were preserved based on hypergeometric testing.

### 2.6. The Integrated Analysis of Transcriptomic and Metabolomic Data

For the integration analysis of genes and metabolites, KEGG pathway analysis was initially used to obtain the links between gene transcripts and metabolites in metabolic pathways. Pearson correlation coefficients were used to calculate the association of metabolic and transcriptomic data between the two groups. Differentially expressed Unigenes were ordered according to *p*-values, and significantly accumulated metabolites were inverted based on VIP values. The top 50 genes and metabolites were selected for heatmap plotting by R packages.

### 2.7. Data Analysis

To examine the significance of differences between groups, the data were analyzed using the statistical program SPSS 25.0 for one-way ANOVA, the chi-square test, and LSD or Duncan’s multiple comparisons depending on the findings of the chi-square test, respectively (*p* < 0.05 was considered significant). ORIGIN software was used to plot the results, which were presented as mean standard deviation (x ± SD).

## 3. Results

### 3.1. Effect of Hypoxic Stress on Histomorphological Feature and NKA Activity in P. akamusi

Compared with the control group (Figure 1A), the *P. akamusi* staining in the hypoxic group was darker and tighter (Figure 1C). The tissue cells in the hypoxic group were haphazardly distributed with different cell morphologies, significantly smaller in size than the control group, and displayed a tighter state (Figure 1D). In contrast, tissue cells in the control group were neatly arranged, maintaining normal cell morphology and volume (Figure 1B). The tissue cells in the hypoxic group tended to be rounded with blue–black chromatin and some of the cell membranes were crinkled and invaginated, indicating apoptosis (Figure 1F black arrows). Some tissues displayed apoptotic vesicles (Figure 1G red-star-shaped marker cells). Under hypoxic stress, a significant number of *P. akamusi* tissues exhibited morphological characteristics of apoptosis.

The hypoxia and control groups showed similar trends in NKA activity over time. The NKA activity of the hypoxic group was significantly higher than that of the control group (Figure 2). The NKA activity of the control group displayed an overall trend of increasing and then fluctuating downward, with significant changes. Specifically, it increased significantly at 12 h, decreased significantly, increased significantly again at 48 h, and decreased significantly again at 96 h with activity similar to the initial value. In the hypoxic group, NKA activity showed a fluctuating upward trend followed by a downward trend with significant changes. Specifically, it increased significantly at 12 h, decreased significantly at 24 h, then increased significantly and reached a maximum at 72 h, and decreased significantly and reached a minimum at 96 h, with activity similar to the initial value.

### 3.2. Effect of Hypoxic Stress on Energy Metabolism in P. akamusi

The glycogen content in the control group was significantly higher than that in the hypoxic group during hypoxic stress. The glycogen content in the control group tended to decrease overall, with significant decreases at 12 h and 96 h and reaching a minimum value at 96 h, which was significantly lower than the initial value. In the hypoxic group, there was a decreasing trend followed by an increasing trend with a significant change. Its content reached its lowest value at 48 h and was significantly lower than the initial value at 96 h (Figure 3A). LDH activity was significantly higher in the hypoxic group than in the control group after 48 h of hypoxic stress. LDH activity in the control group decreased significantly at 24 h, increased significantly at 48 h, and reached a maximum value, then decreased significantly and reached a minimum value at 96 h. This value was significantly lower than the initial value. In the hypoxic group, LDH activity showed a fluctuating upward trend. Its activity increased significantly at 24 h, but decreased significantly at 72 h, then increased significantly and reached a maximum at 96 h, and the value was about 3.5 times higher than the initial value (Figure 3B). Except for at 96 h, the ADH activity of the hypoxic group was significantly higher than the control group at different stress times. Overall, the ADH activity of the control group showed fluctuating but non-significant changes. The ADH activity of the hypoxic group showed a fluctuating decreasing trend. At 48 h of hypoxic stress, the ADH activity continued to decrease. However, at 72 h, the ADH activity increased significantly, then decreased significantly and reached the minimum value at 96 h, and the value was significantly lower than the initial value (Figure 3C).

### 3.3. Effect of Hypoxic Stress on Antioxidant Enzyme Activity in P. akamusi

During hypoxic stress, the SOD activity of the control group was significantly higher than that of the hypoxic group except for at 0 h (Figure 4A). The SOD activity of the control group displayed an overall fluctuating trend with significant change. Specifically, it increased significantly at 12 h, decreased significantly, and reached a low value at 48 h, then increased significantly and reached a maximum value at 96 h, which was significantly higher than the initial value. In contrast, the SOD activity of the hypoxic group showed a trend of increasing and then decreasing, with a significant decrease at 24 h and reaching its lowest value, followed by a significant increase at 72 h and stabilization. However, at 96 h, the SOD activity was significantly lower than the initial value. The CAT activity of the hypoxic group was significantly higher than that of the control group at different stress times (Figure 4B). In the control group, CAT activity increased significantly at 12 h and reached a maximum value, then decreased significantly and reached a minimum value at 48 h, but increased significantly at 72 h and then stabilized. CAT activity was similar to the initial value at 96 h.

### 3.4. GO Enrichment and KEGG Enrichment Analysis of All Differentially Expressed Genes

GO annotation enrichment was performed to analyze differentially expressed genes in *P. akamusi* under hypoxic stress (Figure 5). The results revealed significant enrichment of metabolic and catabolic processes involved in the lipid metabolic process, carbohydrate derivative metabolic process, and trehalose metabolic process from a biological process and ontology perspective. In terms of a molecular function perspective, catalytic activity, oxidoreductase activity, transferase activity, transmembrane receptor protein tyrosine kinase activity, carbohydrate kinase activity, and trehalose activity were significantly enriched. For cellular components, membrane-related components, such as the vesicle membrane, endomembrane system, and extracellular region, were significantly enriched.

KEGG annotation enrichment was used to analyze the differentially expressed genes in *P. akamusi* under hypoxic stress (Figure 6). Several pathways were significantly altered during hypoxic stress, including metabolic pathways, drug metabolism pathways, pyrimidine metabolism, porphyrin and chlorophyll metabolism pathways, ABC transporters, and cytochrome p450-related pathways. The differentially expressed genes in these pathways include ATP-binding cassette, subfamily C (CFTR/MRP), member 4 (ABCC4) ATP-binding cassette, subfamily G (WHITE), member 1 (ABCG1), ATP-binding cassette, subfamily G (WHITE), member 4 (ABCG4), dimethylaniline monooxygenase, (N-oxide forming)/hypotaurine monooxygenase (FMO), glucuronosyl transferase (UGT), etc. (Appendix A).

### 3.5. KEGG Enrichment Analysis of Differentially Accumulated Metabolites

KEGG annotation enrichment was used to analyze the differentially accumulated metabolites in *P. akamusi* related to acute hypoxic stress (Figure 7). The analysis indicated that the significantly enriched pathways (q < 0.05) were mainly associated with amino acid metabolism and synapse, including “Alanine, aspartate and glutamate metabolism”, “Biosynthesis of amino acids”, “Arginine biosynthesis”, “D-Glutamine and D-glutamate metabolism”, “Synaptic vesicle cycle”, “Aminoacyl-tRNA biosynthesis”, “GABAergic synapse”, and “Glutamatergic synapse”. Additionally, “Metabolic pathway”, “ABC transporters”, “Microbial metabolism in diverse environments”, “Glyoxylate, dicarboxylate metabolism”, etc., were also significantly enriched. The differentially accumulated metabolites associated with these pathways included Glutamate, Protoheme, 2,4,5-Trichlorophenol, L-Glutamic acid, 2-Oxoglutarate, and L-Glutamine (Appendix A).

### 3.6. The Integrative Analysis of Transcriptome and Metabolome for P. akamusi in Response to Acute Hypoxia

To screen the associated genes and metabolites, we further performed an integrative analysis of the transcriptome and metabolome using the correlation coefficient model. The Pearson coefficient of gene expression and metabolite abundance was calculated to evaluate the correlation between genes and metabolites, using the differential genes and the differential metabolites in the top 50. The correlation matrix of the heat map revealed the positive (red) and negative (blue) correlations between metabolites and genes, as shown in Figure 8. The correlation data for the top 50 metabolites and genes are displayed in Appendix A. The integrative analysis of the transcriptome and metabolome revealed metabolites associated with the ABC transporter pathway, including “Carboxylic acids and derivatives” and “Biotin and derivatives”. More metabolites associated with microbial metabolism in the diverse environments pathway include “Hydroxy acids and derivatives”, “Keto acids and derivatives”, “Benzene and substituted derivatives”, “Benzene and substituted derivatives”, etc. Pathways related to energy metabolism include protein digestion and absorption, glycolysis/gluconeogenesis, and citrate cycle (TCA cycle), with corresponding metabolites such as “Carboxylic acids and derivatives”, “Indoles and derivatives” and “Organoids”, “Indoles and derivatives”, “Organooxygen compounds”, etc. In addition, Carboxylic acids and derivatives, Keto acids and derivatives, and Dihydrofurans were found in the alcoholism and HIF-1 signaling pathway, and all these metabolites were significantly related to the top 50 genes.

## 4. Discussion

*Propsilocerus akamusi* can tolerate severe hypoxia and become the dominant species in eutrophic lakes, yet the mechanism of this hypoxia tolerance has not been determined. In addition, single omics cannot systematically explain the rapid biochemical reactions and metabolic changes that are expected for acute hypoxia. Thus, we combined physiological indicators and histomorphology observations with metabolome–transcriptome analysis to comprehensively assess the histomorphological features, energy metabolism mechanism, and antioxidant mechanism of *P. akamusi* larvae exposed to acute hypoxia. As well as most predictions being confirmed, additional pathways related to the energy metabolism of the fatty acid, protein, trehalose, and glyoxylate cycles were discovered as well. Furthermore, acute hypoxic stress decreases rather than increases superoxide dismutase (SOD) activity, and we also found that HIF-1 plays an important role in promoting glycogenolysis in acute hypoxic conditions [50,51].

Hypoxic stress significantly altered histomorphological features and NKA activity to result in apoptosis [52]. Meanwhile, the differentially expressed genes (DEGs) associated with the transmembrane transport and vesical membrane were significantly enriched. These results confirmed that cell membrane permeability and osmotic pressure are altered under hypoxic stress, which in turn leads to cell membrane shrinkage and cell volume reduction. The apoptotic mechanism in shrimp gills, hepatopancreas, and hemocytes in response to hypoxia has also been studied [20,23,53]. In the present study, the activity of NKA was promoted and showed a fluctuating upward trend in hypoxic stress. However, the activity of NKA in the previous study was significantly inhibited and found a significant downward trend in crab and clam [26,54]. The reason for the different study results may be that acute hypoxia significantly inhibited O_2_^−^ production, which may put the ROS level in an “optimal redox potential range” and thus promote the activity of NKA [55]. Furthermore, cellular ion balance was altered by promoting NKA activity, which in turn altered tissue cell permeability, increased cytoplasmic density, and led to cellular tightening [56]. The apoptotic vesicles formed were eventually phagocytosed by specialized phagocytes, which may be the reason that the NKA activity decreased significantly under hypoxic stress for 72 h. Therefore, apoptosis may be an important factor in the ability of *P. akamusi* to tolerate acute hypoxic conditions for long periods. In addition, changes in experimental temperature may be the main reason for the tendency of NKA activity to show fluctuating changes. This is also a shortcoming of the present study.

To evaluate the effects of hypoxia on energy metabolism in *P. akamusi*, various parameters were measured. Hypoxic conditions resulted in a significant reduction in glycogen concentrations, indicating a decrease in energy stores [57]. Significant enrichment of HIF-1-related metabolite suggests that HIF-1 plays an important role in promoting glycogenolysis to produce more ATP for Chironomidae to survive under acute hypoxia [50,51]. Additionally, metabolites and DEGs associated with lipids, trehalose, protein digestion and absorption, and the glyoxylate cycle, including ATP-binding cassette, subfamily C (CFTR/MRP), member 4 (ABCC4) ATP-binding cassette, and L-Glutamine, were also significantly enriched, which suggested that they also play an important role in energy metabolic processes under hypoxic stress [58]. The rate of glycogen degradation was slow in acute hypoxic conditions, and the longer-term hypoxia caused a further suppression of the energy metabolism in Chironomus larvae, which may be the reason glycogen no longer decreased after 48 h [28,59,60,61]. Lower glycogen content might be closely related to the increase in LDH activities in the hypoxia group because *P. akamusi*, like *Chaoborus crystallinus* and other macroinvertebrates, might also enhance glycogen decomposition to cope with hypoxia [16,62]. In a previous study, ethanol was thought to be the sole end product of glycogen degradation in *Chironomus riparius* larvae under hypoxic stress [14,15]. This study not only monitored ethanol but also found that ADH activity significantly increased after 48 h of hypoxic stress. However, this is evidence that differentially accumulated metabolites of the microbial metabolism and oxidative pathway, including “Hydroxy acids and derivatives”, “Keto acids and derivatives”, and “Carboxylic acids and derivatives”, were significantly enriched. Animal cells did not have pyruvate decarboxylase, indicating that ethanol was derived from the anaerobic metabolism of the symbiodinium, not from *P. akamusi* larvae [63]. In conclusion, *P. akamusi* coped with energy deficiency and ethanol toxicity under acute hypoxic stress by promoting glycogenolysis and increasing LDH and ADH activities. The potential of ADH and LDH as biomarkers for quick evaluation of hypoxia stress on Chironomidae is supported by the substantial positive connection between hypoxia duration and these enzymes. Our findings provide a new perspective on the etiology of these species by proving that hypoxia is a stressful situation that causes the organism to enter a compensatory response.

The main contribution of acute hypoxia stress to *P. akamusi* may be oxidative stress. This study revealed that extreme hypoxia (DO 0.39 mg O_2_ L^−1^) significantly inhibited the generation of O_2_^−^ and, therefore, limited the activity of the SOD [64]. However, the result was contrary to our prediction. According to prior research, mild hypoxia (DO 4 mg O_2_ L^−1^) increased SOD activity, while severe hypoxia (DO 2 mg O_2_ L^−1^) had no discernible impact on SOD [65,66]. However, SOD activity was suppressed in an anoxic environment for 8 h, which is consistent with the findings of our experiment [67]. In essence, the DO concentration and duration of hypoxia affected the level of SOD [13]. In contrast to SOD, the oxidative stress biomarker CAT was significantly promoted in the acute hypoxia exposure group compared to the control group, and similar results were observed in crab and shrimp [68,69,70]. The reasons for these results may be that lactate produced by *P. akamusi* can be used as a substrate to generate H_2_O_2_ catalyzed by L-α-hydroxy acid oxidases, and ethanol, produced by symbiodinium, also stimulates the function of peroxidase in CAT [64]. Elevated H_2_O_2_ and ethanol levels may result in cell apoptosis in *P. akamusi* [71,72,73,74,75,76]. Our results showing that DEGs associated with oxidoreductase and metabolites associated with the alcoholism pathway were significantly enriched also verified the exploration. In addition, significant enrichment of cytochrome P450 in DEGs, including (N-oxide forming)/hypotaurine monooxygenase (FMO) and glucuronosyl transferase (UGT), indicated that cytochrome P450 also played an important role in promoting ethanol metabolism [77]. In the actual freshwater ecosystem, dissolved oxygen (DO) concentration fluctuations due to anthropogenic pressure and global warming are notable on spatial and temporal scales [5,78,79,80]. However, the DO concentration was the same in this study, and thus future investigations may require different combinations of DO concentration. In addition, this study does not leave a lot of room for generalizing about the mechanisms of hypoxia tolerance, since it is a single-species and single-population study. Therefore, in future studies, we should select more than two species for comparative analyses.

## 5. Conclusions

We combined histomorphological features and biomarkers with the transcriptome and metabolome to answer the question of why *P. akamusi* can tolerate acute hypoxic stress for long periods. Our results confirmed that the altered cell permeability and apoptosis, which was induced by H_2_O_2_ and ethanol, was regulated by NKA, DEGs, and differentially accumulated metabolites. Research results indicated that lipids, protein, and trehalose, as well as glycogen, are also critical in energy metabolism under acute hypoxic stress. Lactate was thought to be the end product of glycogen degradation, and HIF-1 plays an important role in promoting glycogenolysis to produce more ATP for *P. akamusi* to cope with acute hypoxic conditions. In addition, there may be a glyoxylate cycle in *P. akamusi* or symbiodinium, which converts fatty acids into carbohydrates to cope with insufficient energy. Ethanol produced by the anaerobic respiration of symbiodinium was catalyzed by ADH, CAT, and cytochrome P450 to weaken its harm to the organism, and its content also reflects the level of hypoxia in the environment. Although severe hypoxia inhibited SOD activity, lactate, and ethanol promoted CAT activity. These results suggest that some parameters related to energy metabolism, antioxidant enzyme activities, and histomorphological features may be used as biomarkers of eutrophic lakes in *Chironomus riparius* larvae. The study also provides a scientific reference for assessing toxicity and favoring policies to reduce their impact on the environment.

## Figures and Tables

**Figure 1 insects-14-00800-f001:**
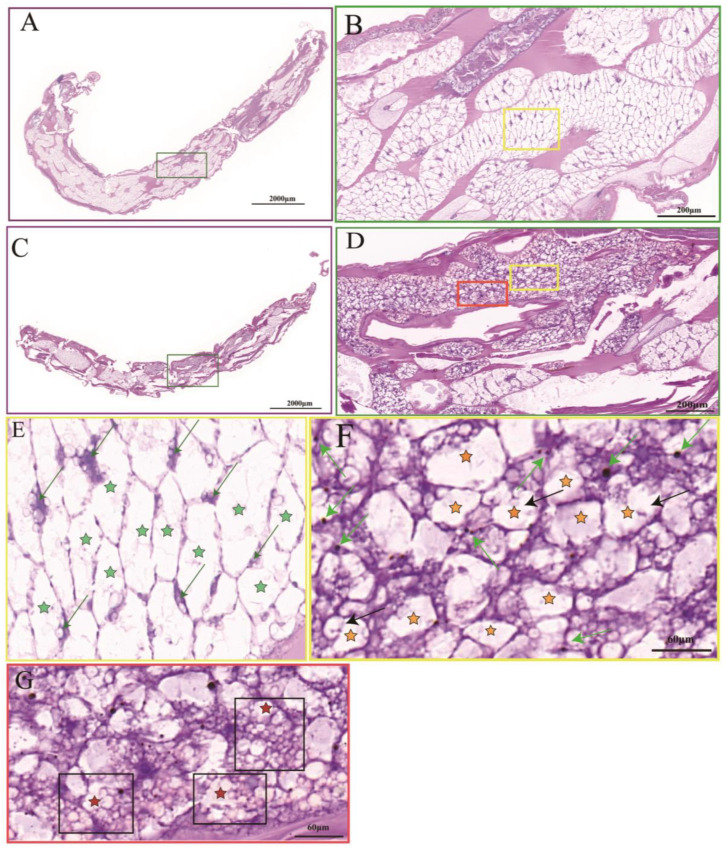
Histomorphological feature of the *P. akamusi*. Note: The control group includes (**A**) (upper; 3×), (**B**) (upper; 30×), and (**E**) (upper; 90×); the hypoxic group includes (**C**) (upper; 3×), (**D**) (upper; 30×), (**F**) (upper; 90×), and (**G**) (upper; 90×). Bright green arrow: chromatin; black arrows (**F**): the cell membranes were crinkled and invaginated; green stars (**E**): normal cell; dark orange stars (**F**): cells exhibiting a crinkled and invaginated state; red stars (**G**): cells producing apoptotic vesicles. (**E**,**F**) represent the tissue cells in the yellow boxes in (**B**,**D**). (**G**) represents the tissue cells with red boxes in (**D**).

**Figure 2 insects-14-00800-f002:**
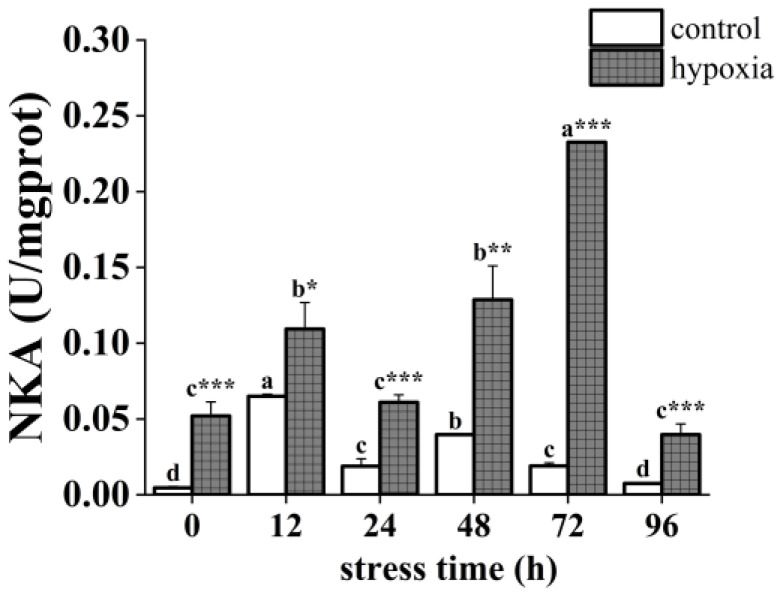
Changes in NKA activity of the *P. akamusi* under hypoxic stress. Note: Different letter superscripts indicate significant differences within the control or hypoxic group in different stress times (*p* < 0.05). * indicates a significant difference between groups at the same time point, * indicates *p* < 0.05, ** indicates *p* < 0.01, *** indicates *p* < 0.001.

**Figure 3 insects-14-00800-f003:**
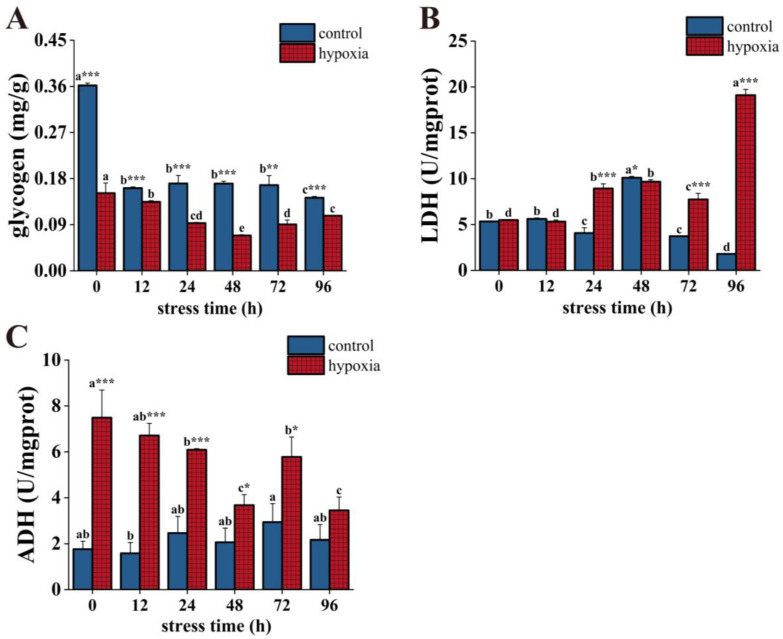
Changes in glycogen content (**A**), ADH activity (**B**), and LDH activity (**C**) of *P. akamusi* under hypoxic stress. Note: Different letter superscripts indicate significant differences within the control or hypoxic group in different stress times (*p* < 0.05). * indicates a significant difference between groups at the same time point, * indicates *p* < 0.05, ** indicates *p* < 0.01, *** indicates *p* < 0.001.

**Figure 4 insects-14-00800-f004:**
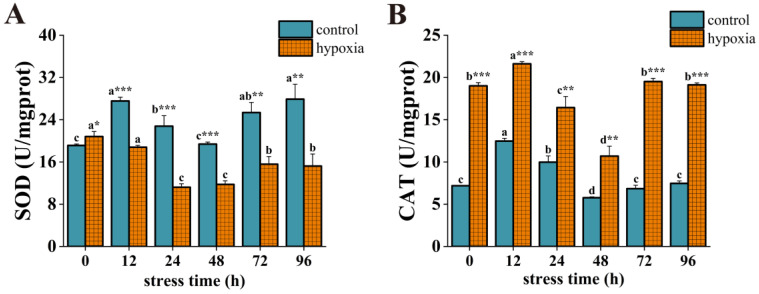
Changes in SOD activity (**A**) and CAT activity (**B**) of *P. akamusi* under hypoxic stress. Note: Different letter superscripts indicate significant differences within the control or hypoxic group in different stress times (*p* < 0.05). * indicates a significant difference between groups at the same time point, * indicates *p* < 0.05, ** indicates *p* < 0.01, *** indicates *p* < 0.001.

**Figure 5 insects-14-00800-f005:**
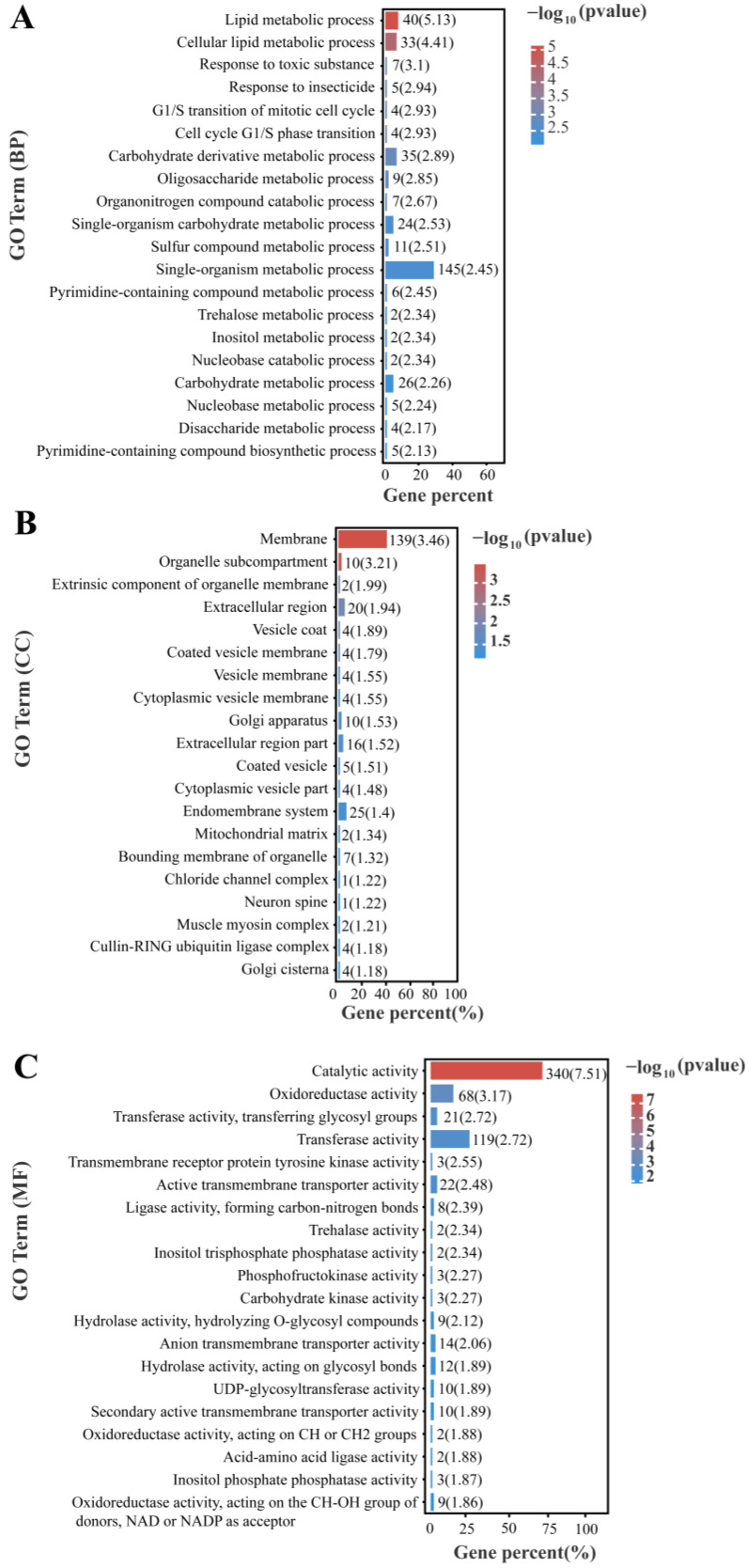
Top 20 GO terms enrichment ((**A**) biological processes, (**B**) cellular component, and (**C**) molecular function) of all DEGs under hypoxia stress.

**Figure 6 insects-14-00800-f006:**
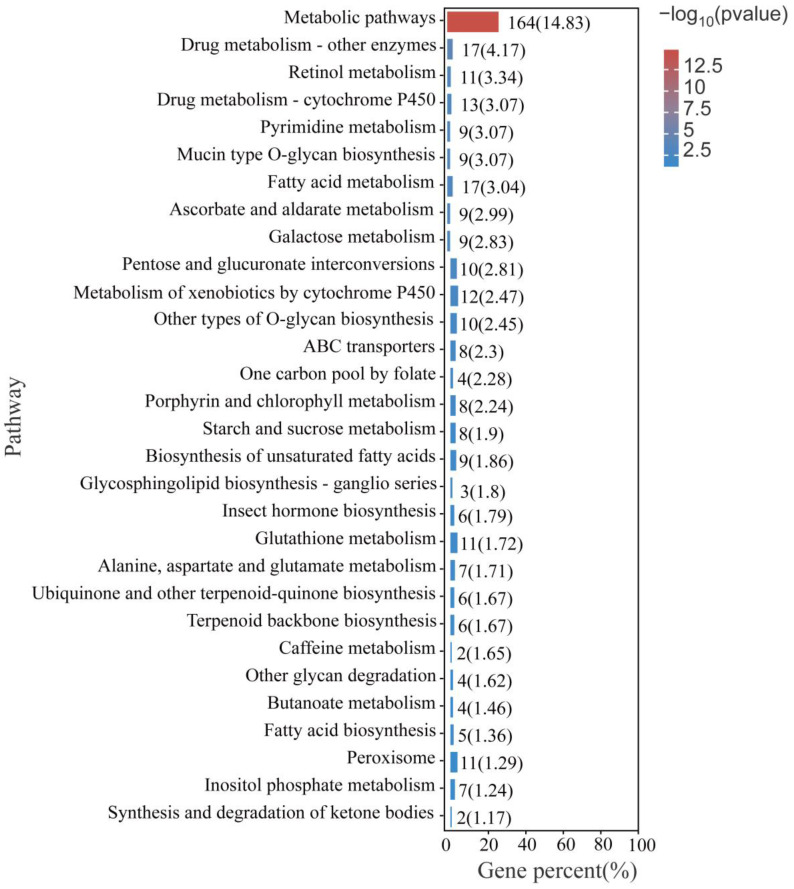
Top 20 of KEGG enrichment analysis of all DEGs.

**Figure 7 insects-14-00800-f007:**
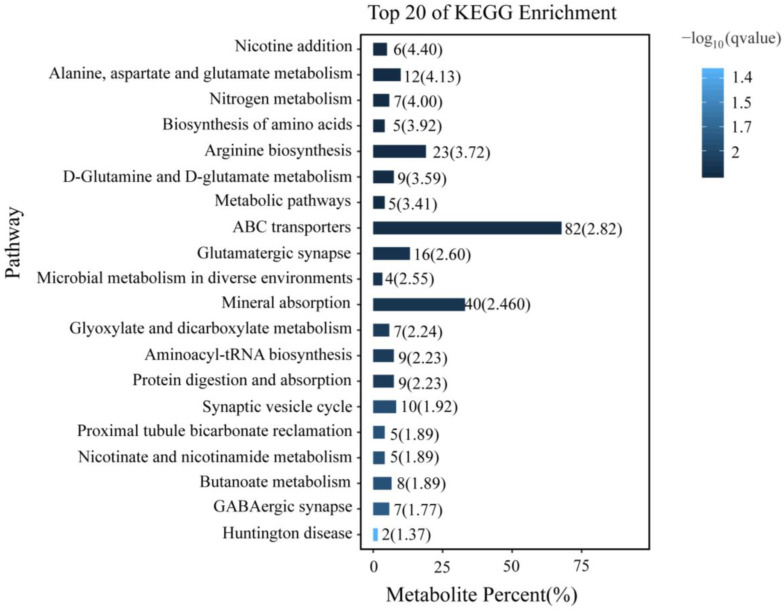
Top 20 of KEGG enrichment analysis of differentially accumulated metabolites.

**Figure 8 insects-14-00800-f008:**
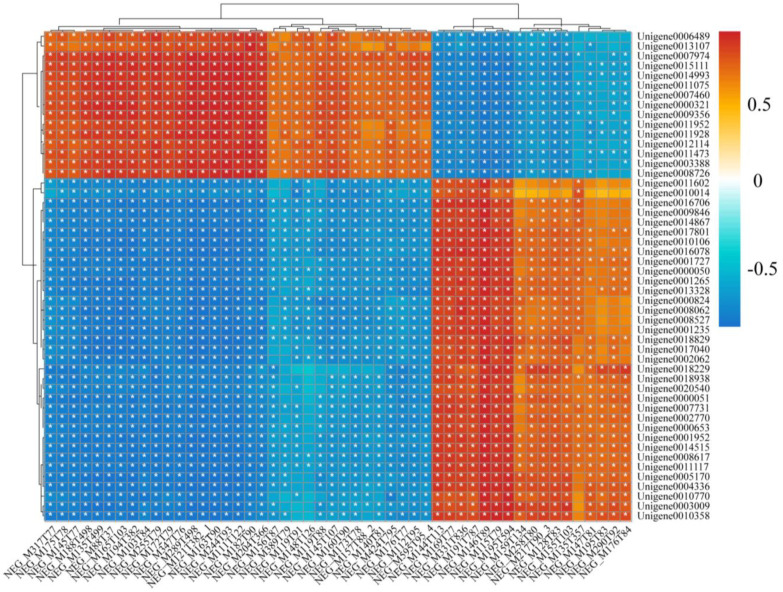
The heat plot of the correlations between metabolites (rows) and genes (columns) in the top 50. The red and blue colors show the positive and negative correlations between transcriptomics and metabolomics data. * indicates *p* < 0.05.

## Data Availability

The data presented in this study are available on request from all authors.

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
