# Peer review of "The Multifaceted Effects of Short-Term Acute Hypoxia Stress: Insights into the Tolerance Mechanism of Propsilocerus akamusi (Diptera: Chironomidae)"

_insects, 2023, doi:10.3390/insects14100800_

Round 1

Reviewer 1 Report

The manuscript focuses on the response of Propsilocerus akamusi, a species of Chironomidae, to acute hypoxic stress. The study uses multi-omics analysis, including transcriptome and metabolome, in combination with histomorphological characteristics and physiological indicators. The research aims to understand how P. akamusi survives prolonged periods of acute hypoxia and how it becomes a dominant species in eutrophic lakes. The manuscript is generally well written, technically sound and conclusions derived from the results are reasonable. However, I feel that there are still some shortcomings in the manuscript that need to be addressed before acceptance. I detail these below as major and minor comments.

Major Comments

1. Clarity of the introduction: The introduction could be improved by providing more context and background information on the importance of Chironomidae in freshwater ecosystems, their ecological role, and the significance of understanding their response to hypoxic stress. This would help readers better understand the motivation for the study and its potential implications.

2. Ecological significance: The article highlights various changes in enzyme activities and metabolites under hypoxic stress, but the ecological significance of these changes for the survival and dominance of P. akamusi in eutrophic lakes could be further explored and discussed.

3. Clarity of writing: The article could benefit from clearer and more concise writing. Some sentences and sections are lengthy and difficult to follow. Improved clarity and organization of the text would enhance the overall readability of the article.

4. Limitations and future directions: The article does not discuss the limitations of the study and potential future directions for research. Acknowledging the limitations of the research and suggesting areas for further investigation would strengthen the conclusions and broaden the study's impact.

Minor Comments

1.       The use of proper punctuation and formatting, such as consistent use of commas and spaces between numbers and units, should be ensured for improved readability.

2.       Line 407-573: Some references in the text lack proper citations, and their citation style is inconsistent. The authors should ensure all references are accurately cited and formatted according to the journal's guidelines.

3.       In the Abstract:

Line 47: Please change “P akamusi” to italics.

Line 43-44: "the energy metabolism of the fatty acid, protein, trehalose, and glyoxylate cycles is also included" could be rephrased to "the involvement of energy metabolism pathways, including fatty acid, protein, trehalose, and glyoxylate cycles."

4.       In the Introduction:

Line 55: "Hypoxic zones have been rapidly expanding" should specify the time frame of the expansion. For example, "Hypoxic zones have been rapidly expanding over the past two decades" or mention the specific time period.

Line 56: "as a result of rising nutrient loadings and global warming" could be clarified by providing references to recent studies supporting this claim.

Line:56-57: "However, acute hypoxia leads to higher mortality rates in numerous freshwater species, especially macroinvertebrates, which are crucial to the provision of many ecosystem functions" should specify the impact of acute hypoxia on specific macroinvertebrate species and their associated ecosystem functions.

5.       In the Materials and Methods:

Line 109: "Hamilton tap water" should be clarified or replaced with the appropriate term to describe the specific water source.

Line 115: "The oxygen concentration in the control group was not less than 8.0mg/L" should be changed to "The oxygen concentration in the control group was maintained at 8.0 mg/L or higher" for clarity.

Line 106: Provide more information about the "Surber net" used to collect P. akamusi larvae, including its mesh size or any specific details related to its use.

Line 148: In the "Transcriptome analysis" section, specify the sequencing platform used for the RNA sequencing (e.g., Illumina HiSeq or NovaSeq).

Line 165-171 In the "Metabolome analysis" section, provide details on the mass spectrometry platform used for metabolite identification (e.g., LC-MS or GC-MS).

6.       In the Results:

Line 190: In Figure 1, Show what the yellow box and the red box represent.

Line 220: In Figure 3, provide error bars or standard deviation to represent the variability of data points.

Line 242: In Figure 4, clarify the y-axis label to indicate the units used for SOD and CAT activity measurements.

Line 255: In the text, provide a more detailed explanation of the results obtained in the GO and KEGG enrichment analyses.

7.       In the Discussion:

Line 320: In the discussion of NKA activity, clarify the potential reasons for the fluctuating trend observed under hypoxic stress.

Line 355-364: Provide more context and relevant studies to support the findings related to oxidative stress and the activities of SOD and CAT in response to hypoxic stress and Please replace -1 with a superscript.

Line 331-333: In the discussion of energy metabolism, specify the specific metabolites associated with different pathways and their potential roles in P. akamusi's response to hypoxia.

Line 373-374: Add the limitations of the study and potential sources of variability in the results.

Minor editing of English language required.

Author Response

Thank you for your comments. We have made the necessary revisions to the manuscript based on your suggestions, and we have directly addressed the specific errors within the manuscript.

Reviewer 2 Report

Please, improve all technical aspects of submitted manuscript. Not only that some insect or other species' Latin names are not written in Italic, but also there are many spaces that need to be included after words and number of cited references in brackets, as well as there are plenty of words with capital letters in the middle of the sentence. Moreover, some of the sentences are missing full stops! It is unbelievable that colleagues were unable to proofread the manuscript before final submission to the journal! 

Extensive language editing is needed, both for style and grammar. Please, do not start sentence with ,,And", do not use ,,we did this, we showed that"... Simple summary and abstract are full of language errors and are quite incomprehensible to readers, even though they are the most important parts for a prospective reader's attention!

Also, not less important, please bear in mind subscript/superscript when writing superoxide anion radical, hydrogen peroxide and other chemical symbols...it is unacceptable to have such errors in the manuscript!

Extensive language and technical editing needed prior to further revision of the manuscript!

Author Response

(The authors gave the same response as above.)

Reviewer 3 Report

Title:  The Multifaceted Effects of Short-Term Acute Hypoxia Stress:  Insights into the tolerance mechanism of Propsilocerus akamusi  (Diptera: Chironomidae) 

The authors explore the histological, transcriptomic, and metabolomic responses of a chironomid to hypoxia. The premise of the paper is fairly sound; using multiple metrics to understand the effects of a stressor on animal physiology can be quite interesting. However, I found the paper to be unfortunately very difficult to read. Numerous grammatical issues prevented me from fully understanding and evaluating the research. My first and major suggestion would be for the authors to check grammar and sentence structure in the manuscript. Below are more comments which I hope will be helpful to authors as they revise:

General:

I would urge the authors to check the grammar, sentence structure, and flow of ideas in their manuscript. Some of it is incorrect and requires fixing. I am proving but a few exmples below, but there are many, many more throughout the manuscript:

Ln 62: The tenses in the sentences should be correct. Change “altered” to “alters”

Ln 44 and 97: You cannot start a sentences with “And”.

Ln 316 and 335-336: “significantly enrichment” is grammatically incorrect. It should be “significantly enriched” or “significant enrichment”. 

Ln 368: “Alcoholism pathway” is wrong,; it should be something else.. potentially oxidative or non-oxidative pathways?

Methods and Results:

I don’t really like the fact that this is a single species and single population study — it doesn’t leave a lot of room for generalizing about the mechanisms of hypoxia tolerance, which is a very interesting subject in ecological physiology. I realize it might be too late, but if the authors are able to add another population of these chironomids, it might strengthen the interpretations.

The authors are looking at short-term tolerance to hypoxia, yet they have different time points, from 0 - 96h. I did not understand the reasoning behind using so many time points. The authors should include a thoughtful explanation for why they used numerous time points.

I am not familiar with histological analyses, so I did not know how appropriate the methods were. However, it was not easy for me to really see what the arrows were pointing to and how they were different between the control and hypoxia treatment groups. I would request the authors deepen their explanations of why histology is a good method for finding insect reposes to hypoxia, and what exactly the reader should see in the Fig 1. Note that this is a general audience for the paper, so explanations should be helpful and detailed. Further, the arrows are bright green, not blue, so it took me sometime to realize that the authors were talking about the green arrows.

Figure 3B, the two bars at the 0 time point don’t look significantly different, yet there is a star on the second bar. Can the authors recheck this?

Ln 290 — the subtitle mentions clams, not midges. 

Discussion:

I feel the discussion lacks the depth that most published discussions need. I would first like to see the authors briefly reiterate the motivation for their study and potentially some of the predictions. Then I would like to see detailed and thoughtful discussion of each result and an appropriate contextualization of the result in the published literature. In addition, a number of caveats need to be discussed. For example, GO terms are notoriously over-interpreted by researchers and I feel this study is no exception. GO terms are largely developed from human genes and functions and do not necessarily provide ecologically relevant information for animals such as insects. It’s important to state these caveats clearly.

The authors should also mention what could be done differently and what future researchers might be able to do after this study. 

Without some of these basic elements, I feel the Discussion is somewhat shallow and dissatisfying.

If the authors are able to make substantial revisions to their presentation of this research, it will more likely be acceptable for publication. 

Please see my comments above. Extensive editing is required.

Author Response

(The authors gave the same response as above.)

Round 2

Reviewer 2 Report

I am satisfied with the presented/improved version of the manuscript.